# Inositol Polyphosphate-Based Compounds as Inhibitors of Phosphoinositide 3-Kinase-Dependent Signaling

**DOI:** 10.3390/ijms21197198

**Published:** 2020-09-29

**Authors:** Tania Maffucci, Marco Falasca

**Affiliations:** 1Centre for Cell Biology and Cutaneous Research, Blizard Institute, Barts and The London School of Medicine and Dentistry, Queen Mary University of London, London E1 2AT, UK; 2Metabolic Signalling Group, School of Pharmacy and Biomedical Sciences, Curtin Health Innovation Research Institute, Curtin University, Perth, WA 6102, Australia

**Keywords:** inositol polyphosphates, inositol 1,3,4,5,6-pentakisphosphate, phosphoinositides, phosphoinositide 3-kinase, pleckstrin homology domain

## Abstract

Signaling pathways regulated by the phosphoinositide 3-kinase (PI3K) enzymes have a well-established role in cancer development and progression. Over the past 30 years, the therapeutic potential of targeting this pathway has been well recognized, and this has led to the development of a multitude of drugs, some of which have progressed into clinical trials, with few of them currently approved for use in specific cancer settings. While many inhibitors compete with ATP, hence preventing the catalytic activity of the kinases directly, a deep understanding of the mechanisms of PI3K-dependent activation of its downstream effectors led to the development of additional strategies to prevent the initiation of this signaling pathway. This review summarizes previously published studies that led to the identification of inositol polyphosphates as promising parent molecules to design novel inhibitors of PI3K-dependent signals. We focus our attention on the inhibition of protein–membrane interactions mediated by binding of pleckstrin homology domains and phosphoinositides that we proposed 20 years ago as a novel therapeutic strategy.

## 1. Phosphoinositides and Protein Binding Domains

Phosphoinositides consist of a water-soluble inositol head linked to a diacylglycerol by a phosphodiester linkage (Figure 1) [1]. Although nine isomeric forms of inositol exist, myo-inositol is the most commonly used in nature [2]. The majority of phosphoinositides possess the same acyl chains (stearoyl in the 1-position and arachidonoyl in the 2-position) and, historically, little attention was paid to determining the physiological relevance of such a specific enrichment [3]. Recently, thanks to improved methodologies, there is an increased interest to understand the importance of differential acyl chains within phosphoinositides. Traditionally, however, phosphoinositides have been classified on the basis of the specific phosphorylation of their myo-inositol head (Figure 1).

While the precursor and most abundant member of the family, Phosphatidylinositol, is unphosphorylated (Figure 1a), differential phosphorylation of the hydroxyls at the 3-, 4-, and 5-position of myo-inositol leads to seven derivatives (Figure 2), all of which have been detected in all higher eukaryotes [4].

Intracellular localization of specific kinases and phosphatases and/or their relocation upon cellular stimulation regulate synthesis and degradation of localized pools of phosphoinositides. Such a spatial and often temporal regulation makes phosphoinositides, despite being only minor components of cellular membranes, key modulators of several physiological processes, mostly by controlling compartmentalization and, often, activation of diverse cellular proteins [4]. This is achieved through their binding to specific domains within their target proteins, which, in turn, retains/relocates these proteins to the specific cellular compartment where their function is required. As these protein domains bind to the soluble head group, the distinct phosphorylation pattern within the inositol ring can guarantee specificity of interaction between one phosphoinositide and its effector protein. An example of such a selective interaction is provided by proteins that possess Fab1/YOTB/Vac1/EEA1 (FYVE) domains, zinc finger modules that exist as single or tandem repeats and that can specifically bind Phosphatidylinositol 3-phosphate (PtdIns3*P*) [5]. On the other hand, some protein domains do not appear to be specific for one phosphoinositide alone, as in the case of mammalian Phox homology (PX) domains, which have been found to bind PtdIns3*P* but also Phosphatidylinositol 3,5-bisphosphate and Phosphatidylinositol 3,4-bisphosphate (PtdIns(3,4)*P*_2_) [6]. Moreover, there are only few examples of protein domains for which binding to phosphoinositides is sufficient to guarantee a strong membrane interaction of the protein. For example, most FYVE domain-containing proteins require dimerization of these domains to be targeted to PtdIns3*P*-enriched endosomes successfully, although this may be crucial only for a subset but not for all FYVE domains [7]. Similarly, there are only few examples of PX domains able to bind PtdIns3*P* with high affinity, and the majority of them are recruited to the membrane only as part of a multicomplex [4].

Amongst all phosphoinositide-binding domains, pleckstrin homology (PH) domains have received considerable attention since their first identification as a structural protein module of approximately 100 amino acid residues occurring twice in pleckstrin in 1993 [8,9]. Despite a poorly conserved primary structure, all PH domains possess a similar tertiary structure [10]. Although originally PH domains were thought to bind proteins and indeed some protein interactions have been described [11], attention towards them grew when they were recognized as phosphoinositide-binding domains. Only a few PH domains, however, have shown a clear stereospecificity and high binding affinity for phosphoinositides. On the basis of this, almost 20 years ago, we classified them into three groups: (i) PH domains with high affinity for a specific phosphoinositide, (ii) PH domains with low specificity and/or affinity, and (iii) PH domains that bind non-specifically to phosphoinositides [11]. Binding of phosphoinositides to group 1 PH domains is sufficient to relocate/retain the host protein to the membrane. On the other hand, the weak interaction between phosphoinositides and group 2 PH domains requires additional mechanisms to guarantee a stable recruitment of the protein to the membrane [6,11]. We proposed a mechanism of cooperative binding between lipids or lipid and proteins [11]. This mechanism of lipid cooperativity has been subsequently demonstrated [12,13]. More recently, it has been found that the interaction of a PH domain with a membrane requires a localized cluster of phosphoinositide molecules in an anionic lipid-enriched microenvironment [14,15].

An important feature of many PH domains is their ability to bind phosphoinositides as well as their corresponding water-soluble inositol polyphosphates, often with a similar affinity/specificity. This is not common to all phosphoinositide-binding domains; for instance, FYVE domains were reported to bind more strongly to membrane-embedded PtdIns3*P* than to the isolated head group Inositol 1,3-bisphosphate [6,7]. This feature has been crucial for the development of targeted strategies to stop/prevent PH domain/phosphoinositide interactions, as we discuss later.

## 2. Akt PH Domain and PtdIns(3,4,5)*P*_3_: A “Textbook” Example of Phosphoinositide-Dependent Regulation of Signaling Pathways and Cellular Functions

Amongst the seven phosphoinositides, the central role of Phosphatidylinositol 3,4,5-trisphosphate (PtdIns(3,4,5)*P*_3_) in cell signaling was immediately recognized once it emerged that this lipid was almost undetectable in unstimulated normal cells, and it was synthesized very rapidly upon cellular stimulation before being degraded quickly, showing all features of a second messenger. PtdIns (3,4,5)*P*_3_ is one of the three lipid products of a family of enzymes named Phosphoinositide 3-kinases (PI3Ks) because of their ability to catalyze phosphorylation at the inositol 3′ position within selective phosphoinositides [16,17]. Their substrate specificity, together with their structure, has provided the basis for grouping the eight mammalian PI3Ks into three classes [18,19,20]. Specifically, synthesis of PtdIns(3,4,5)*P*_3_ in vivo is mediated by the class I subgroup, dimers of a regulatory protein, and one of four catalytic subunits, namely, p110α, p110β, p110δ, and p110γ [19]. By synthesizing PtdIns(3,4,5)*P*_3_, class I PI3Ks modulate activation of several target proteins, ultimately regulating many cellular functions, including growth, survival, migration, and metabolism [21,22,23].

The most studied and best characterized amongst all proteins activated by class I PI3Ks is the Serine/Threonine kinase Akt [24]. In fact, the class I PI3K/PtdIns(3,4,5)*P*_3_-dependent Akt activation is one of the best-known examples of a phosphoinositide-mediated protein activation, often used as the canonical example of how stimuli (such as growth factors, hormones, etc.) can activate intracellular signaling pathways and functions. Specifically, upon cellular stimulation, class I PI3Ks catalyze the phosphorylation of Phosphatidylinositol 4,5-bisphosphate (PtdIns(4,5)*P*_2_), leading to the synthesis of PtdIns(3,4,5)*P*_3_ at the plasma membrane. PtdIns(3,4,5)*P*_3_ can then bind Akt PH domain and induce translocation of the kinase to the plasma membrane. Once at the membrane, Akt is activated via phosphorylation at its residue Thr308 by 3-phosphoinositide-dependent kinase 1 (PDK1), itself possessing a PtdIns(3,4,5)*P*_3_-binding PH domain. Additional kinases, including the mechanistic target of rapamycin (mTOR) complex 2, further phosphorylate Akt at Ser473, providing full activation of the enzyme. Importantly, binding to PtdIns(3,4,5)*P*_3_ not only localizes Akt to the plasma membrane but it also affects the interaction between its PH and kinase domains [25]. Several studies have demonstrated that such an interaction is crucial for Akt phosphorylation and subsequent dissociation from the plasma membrane, highlighting the importance of this allosteric regulation for Akt activation [25,26]. In particular, it was reported that Akt exists in a complex with PDK1, which is inactive when Akt PH and kinase domains interact, in a so-called “PH-in” conformer [27]. Upon Akt recruitment to the membrane, a conformational change occurs, and in such a “PH-out” conformer, PDK1 can phosphorylate Akt [27].

Once activated, Akt can phosphorylate and activate/inactivate more than 100 substrates, controlling a multitude of cellular functions including cell survival, cell cycle, cell growth, glucose transport, and storage [28]. Considering its ability to regulate a vast array of cellular functions, it is not surprising that Akt activation is tightly regulated, mainly via a strict temporal regulation of PtdIns(3,4,5)*P*_3_ levels. In turn, this is the result of a fine balance between class I PI3K activation and the action of the enzyme Phosphatase and tensin homolog (PTEN), which dephosphorylates PtdIns(3,4,5)*P*_3_ at its 3-position, converting it back to PtdIns(4,5)*P*_2_ (Figure 3) [29]. Disruption of such a balance, due to hyperactivation of class I PI3Ks or deletion/loss of function of PTEN, results in uncontrolled accumulation of PtdIns(3,4,5)*P*_3_ and constant activation of Akt and Akt-dependent pathways and cellular functions. Importantly, another PI3K product, PtdIns(3,4)*P*_2_, is also able to bind to the Akt PH domain. Emerging evidence suggests that this phosphoinositide can have signaling functions independently of PtdIns(3,4,5)*P*_3_, and it has been suggested to control Akt activity throughout the endosomal membranes [30,31]. Interestingly, recent data [32] have indicated that the interaction of calcium with PtdIns(3,4,5)*P*_3_ or PtdIns(3,4)*P*_2_ prevents membrane localization of PH domains of Akt, phospholipase C-δ1 (PLCδ1), and insulin receptor substrate 1 [33].

## 3. PI3Ks, PtdIns(3,4,5)*P*_3_, and Akt in Cancer

Deregulation of the class I PI3Ks/PtdIns(3,4,5)*P*_3_/Akt pathway occurs in several cancer types [19]. Activating mutations of class I PI3Ks, mainly of *PIK3CA*, the gene encoding the catalytic subunit p110α, have been found in several cancer types along with copy number gains or amplification of class I PI3Ks and alteration of the regulatory subunits [34]. Similarly, genetic and epigenetic alterations that result in loss of PTEN function are very common in several cancer types [35]. Additional deregulation normally found in many cancer types would also result in hyperactivation of class I PI3Ks-dependent pathways, such as genetic alterations of receptor tyrosine kinases and other upstream regulators of PI3Ks [35]. Due to its central role in promoting development and progression of several cancer types, the PI3Ks/Akt pathway has been long recognized as an important target to develop novel anti-cancer drugs [20,36,37]. Several inhibitors targeting different components of the pathway have been developed over the years, from drugs targeting all PI3Ks or selective class I PI3K enzymes to drugs targeting Akt or Akt downstream effectors, including mTOR [38]. Overall, however, the success of these drugs has been somewhat disappointing, mainly because of limitations in the doses that can be used to avoid affecting critical physiological functions and because of the development of mechanisms of resistance [39]. Only a small number of these inhibitors are currently approved for use in specific cancer settings, with more clinical trials and additional studies underway [40,41]. Studies to identify novel strategies to block this pathway more efficiently and overcome these limitations are still required.

## 4. Inhibiting Akt by Targeting Its Mechanism of Activation

### 4.1. Preventing Akt Translocation to the Plasma Membrane

While many of the PI3Ks/Akt inhibitors compete with ATP, thus preventing the catalytic activity of the kinases directly, identification of the PH domain-mediated mechanisms of activation paved the road to the development of alternative strategies to inhibit Akt and its signaling pathways. In fact, the first Akt inhibitor to enter clinical trial was perifosine, an alkyl-lysophospholipid (ALP), which was reported to inhibit Akt translocation to the membrane and its subsequent phosphorylation [42]. Other ALPs, such as edelfosine and miltefosine, were also found to be able to inhibit growth factor-induced Akt activation [43]. Surface plasmon resonance spectroscopy studies ruled out a direct binding of perifosine or edelfosine to purified Akt PH domain [44]. On the other hand, it was proposed that perifosine and other ALPs might exert their effects by accumulating at the plasma membrane [45]. It has also been suggested that a decreased content of plasma-membrane cholesterol might be partly responsible for the reduced Akt activation [46].

### 4.2. Targeting Akt PH/Kinase Domain Interaction 

#### 4.2.1. Allosteric Inhibitors

Several allosteric Akt inhibitors have been developed and brought to clinical trial [47]. Their development was mainly driven by the search for more specific drugs that could overcome limitations due to off-target effects of ATP-competitive inhibitors. These drugs act by mainly targeting the interaction between the PH and kinase domains. Examples include ARQ 092, ARQ 751 [48,49], and BAY1125976 [50]. MK-2206 [51] is the most clinically advanced amongst the allosteric inhibitors but has shown limited clinical activity as monotherapy in phase II trials [47,52].

#### 4.2.2. Phosphatidylinositol Ether Lipid Analogues

While allosteric inhibitors mainly target the interaction of Akt PH domain with the kinase domain, an alternative strategy has also been pursued, aiming at interfering with its binding to PtdIns(3,4,5)*P*_3__._ Phosphatidylinositol ether lipid analogues were designed to prevent Akt translocation to the plasma membrane, mainly on the basis of the original demonstration that D-3-deoxy-3-substituted myo-inositols appeared to inhibit growth of some transformed but not wild type NIH 3T3 cells [53]. This led to the development of a series of 1-O-octadecyl-3-deoxy- or 3-hydroxymethyl-phosphatidylinositol ether lipids and related carbonate analogues [54,55,56], as well as a strategy based on the use of D-3-deoxy-phosphatidyl-myo-inositols that cannot be phosphorylated in the 3-position, which would interfere with the correct PtdIns(3,4,5)*P*_3_/PH domain interaction [45]. Indeed, these compounds were shown to bind to Akt PH domain, preventing its translocation to the plasma membrane and activation of the enzyme [57].

#### 4.2.3. Inositol Polyphosphates

As mentioned before, a very important feature of many PH domains is their ability to bind water-soluble inositol polyphosphates, sometimes even with higher affinity than the corresponding phosphoinositides [58]. For instance, we reported that the PH domain of the protein General receptor for phosphoinositides, isoform 1, was able to bind PtdIns(3,4,5)*P*_3_, as well as its corresponding inositol polyphosphate, Inositol 1,3,4,5-tetrakisphosphate (Ins(1,3,4,5)*P*_4_) (Figure 4) [33], consistent with previous studies [59]. Similarly, we reported that PLCδ1 PH domain, which has a strong affinity and specificity for PtdIns(4,5)*P*_2_, also binds the corresponding Inositol 1,4,5-trisphosphate (Ins(1,4,5)*P*_3_) (Figure 4) [33], consistent with previous studies [60]. Similarly, Akt PH domain was found to bind to Inositol 1,3,4,5,6-pentakisphosphate (Ins*P*_5_), and to a slightly lesser extent to Inositol 1,4,5,6-tetrakisphosphate (Ins(1,4,5,6)*P*_4_) [58].

Twenty years ago, we proposed that such a feature could be exploited as an alternative strategy to block the interaction between phosphoinositides and PH domains, as water-soluble inositol polyphosphates could be delivered intracellularly and compete with phosphoinositides for PH domain binding [61]. In particular, we proposed that inositol polyphosphates that could compete with PtdIns(3,4,5)*P*_3_ for its binding to Akt PH domain might block activation of this protein kinase. In our original study, we demonstrated that treatment of COS7 cells with exogenous Ins(1,4,5,6)*P*_4_ was indeed able to inhibit translocation of a Green fluorescent protein (GFP)-tagged Akt PH domain to the plasma membrane [62], providing one of the first pieces of evidence that water-soluble inositol polyphosphates could prevent Akt membrane targeting. Consistent with the inhibition of Akt translocation, exogenous Ins(1,4,5,6)*P*_4_ and Ins*P*_5_ were the only inositol polyphosphates able to inhibit proliferation/growth of breast cancer cell line MCF7 and small cell lung carcinoma (SCLC) cell line H69. No effect was detected when cells were treated with Ins(1,4,5)*P*_3_, Ins(1,3,4,5)*P*_4_, or Inositol 1,2,3,4,5,6-hexakisphosphate (Ins*P*_6_), whereas treatment with inositol 3,4,5,6-tetrakisphosphate (Ins(3,4,5,6)*P*_4_) resulted in ~25% inhibition. These data indicated a specific effect of selective inositol polyphosphates, likely those able to bind to Akt PH domain [62]. As a further confirmation of this hypothesis, Ins(1,4,5,6)*P*_4_ affected growth of the ovarian cancer SKOV-3 cells, which are characterized by constitutive PI3K activation, but not growth of colon carcinoma SW620 cells, which does not depend on PI3K [62]. This original study prompted us to propose a mechanism by which Ins(1,4,5,6)*P*_4_ and Ins*P*_5_ were able to block Akt recruitment to the membrane by binding to its PH domain and competing with PtdIns(3,4,5)*P*_3_.

Subsequent studies further confirmed the anti-cancer properties of Ins*P*_5_. First, we demonstrated that Ins*P*_5_ induced apoptosis of (SCLC)-H69 cells and SKOV-3 cells [63]. Neither Ins*P*_6_, Ins(3,4,5,6)*P*_4_, nor Ins(1,4,5,6)*P*_4_ were able to induce apoptosis SCLC-H69 cells. Importantly, Ins*P*_5_ proved to be as potent as etoposide in SCLC-H69 cells and as cisplatin in SKOV-3 cells. Consistent with our hypothesis of a competition with PtdIns(3,4,5)*P*_3_, Ins*P*_5_, but not Ins(3,4,5,6)*P*_4_ or Ins(1,4,5,6)*P*_4_, inhibited Akt phosphorylation at Ser473 or phosphorylation of the Akt downstream effector Glycogen synthase kinase 3 in SKOV-3 cells. The fact that overexpression of a myristoylated, constitutively active Akt blocked the pro-apoptotic activity of Ins*P*_5_ further confirmed its mechanisms of action [63].

The ability of Ins*P*_5_ to inhibit Akt activation was not limited to cancer cell lines, as this was also observed in PTEN^–/–^ embryonic stem cells [63] and in human umbilical vein endothelial cells stimulated with basic fibroblast growth factor (FGF-2) [64]. In both cases, treatment with other inositol polyphosphates had no effect. Consistent with the inhibition of the PI3K/Akt pathway, Ins*P*_5_ inhibited FGF-2-induced cell survival to the same extent as two Akt inhibitors [64]. This study further revealed that Ins*P*_5__,_ but not other inositol polyphosphates tested, reduced the FGF-2-mediated cell migration, capillary tube formation in vitro, and angiogenic response in vivo. The pro-apoptotic and anti-angiogenic properties of Ins*P*_5_ resulted in strong anti-tumor activity in vivo, confirmed by reduced tumor growth upon treatment with Ins*P*_5_ but not with Ins*P*_6_ in a xenograft model of SKOV-3 cells. Importantly, Ins*P*_5_ reduced tumor growth to the same extent as cisplatin and inhibited Akt activation in vivo, as assessed by analysis of dissected tumors. In this respect, this study provided the first demonstration that exogenous Ins*P*_5_ could inhibit tumor growth and Akt phosphorylation in vivo [64]. Evidence of an endogenous regulation of Akt activation by highly phosphorylated inositol polyphosphates also appeared, as was the case in a study reporting that genetic ablation of the inositol hexakisphosphate kinase IP6K1, one of the enzymes responsible for synthesis of 5-diphosphoinositolpentakisphosphate (5-Ins*P*_7_), resulted in increased Akt signaling in mouse embryonic fibroblasts upon stimulation with insulin-like growth factor 1, as well as in gastrocnemius muscle, epididymal white adipose tissue, and primary hepatocytes in response to acute insulin treatment [65]. The authors further showed that ^3^H-Ins*P*_7_ was able to bind to full-length Akt, while binding to a mutant Akt lacking the PH domain was strongly reduced [65], supporting the conclusion that endogenous Ins*P*_7_ can compete with PtdIns(3,4,5)*P*_3_ for binding to the Akt PH domain.

## 5. Improving the Activity of Ins*P*_5_—Chemical Modifications

### 5.1. Chemical Modifications that Resulted in Inhibition of Additional and Selective Targets

Once we confirmed that the inositol polyphosphates-dependent strategy was successful in blocking Akt activation in vitro and in vivo, we focused our attention on the possibility of improving the anti-cancer activity of Ins*P*_5_ by designing novel synthetic compounds based on the structure of this inositol polyphosphate. Modifications were made on either the 2-*O*-atom or the 5-phosphate, thus maintaining the symmetry of the parent molecule, and this led us to identify the derivative 2-*O*-benzyl-myo-inositol 1,3,4,5,6-pentakisphosphate (2-*O*-Bn-Ins*P*_5_) as the most efficient of those tested (Figure 5) [66].

Characterization of this compound confirmed that it was able to inhibit the platelet-derived growth, factor-induced conformational change of a GFP/Red fluorescent protein-tagged-Akt, resulting in reduced Akt phosphorylation in NIH 3T3 cells overexpressing PDK1 and Akt [67]. Moreover, 2-*O*-Bn-Ins*P*_5_ proved to be more efficient than the parental molecule at reducing Akt phosphorylation, inhibiting cell growth and inducing apoptosis in cell lines that are sensitive to Ins*P*_5_, such as SKOV-3 and SKBR3 cells [66]. Furthermore, 2-*O*-Bn-Ins*P*_5_ was able to inhibit Akt phosphorylation and growth of cell lines normally resistant to Ins*P*_5_, such as prostate cancer PC3 cells and pancreatic cancer ASPC1 cells [66]. More importantly, 2-*O*-Bn-Ins*P*_5_ inhibited growth of PC3 cells in a xenograft model, whereas Ins*P*_5_, used at the same doses, had no effect. The anti-cancer activity in vivo was mirrored by reduced Akt phosphorylation in dissected tumors. This study demonstrated that the addition of a benzyl group to the axial 2-*O* atom of Ins*P*_5_ resulted in enhanced pro-apoptotic and anti-tumor properties [66].

Interestingly, 2-*O*-Bn-Ins*P*_5_ was found to be able to inhibit PDK1 activity in vitro, with an IC_50_ in the low nanomolar range [66]. Further investigation confirmed that 2-*O*-Bn-Ins*P*_5_ was able to bind to PDK1 PH domain with an affinity of 109 ± 44 nM and a 1:1 stoichiometry [67]. These data led us to conclude that 2-*O*-Bn-Ins*P*_5_ was more efficient than its parental molecule, likely because of its ability to target Akt and PDK1 simultaneously and therefore to inhibit additional PI3K/PDK1-dependent, Akt-independent pathways. In this respect, it is noteworthy that we further demonstrated that, alongside Akt inhibition, 2-*O*-Bn-Ins*P*_5_ inhibited the formation of a protein complex between PLCγ1 and PDK1 [67], a complex we had previously reported as being important for breast cancer cell migration and invasion [68]. Specifically, we demonstrated that 2-*O*-Bn-Ins*P*_5_ reduced the epidermal growth factor-induced translocation of both enzymes to the plasma membrane and PLCγ1 activation, resulting in inhibition of cell migration and invasion in a panel of cancer cells as well as metastasis dissemination in a zebrafish embryo model [67].

These studies represented an important step forward in the development of inositol polyphosphate-based drugs. Indeed, one of the main concerns towards the use of inositol polyphosphates is the possibility that they might interfere with additional phosphoinositide-dependent signaling, resulting in non-specific effects. In this respect, our demonstration that 2-*O*-Bn-Ins*P*_5_ was able to inhibit a very specific, PDK1-mediated mechanism of PLCγ1 activation, without interfering with the normal regulation of this enzyme, proved that it is possible to modify the structure of an inositol polyphosphate and to synthesize compounds that can target enzymes in a more selective manner.

### 5.2. Chemical Modifications That Increase Intracellular Delivery

While evidence demonstrating the anti-cancer activity of Ins*P*_5_ accumulated, the crucial question concerning its exact mechanism of action and, specifically, the precise mechanism of its incorporation/cellular uptake, remained to be addressed. Efficient incorporation of inositol polyphosphates, in particular of Ins(1,3,4,5)*P*_4_, was already reported in our original study where starved H69 cells were incubated with ^3^H-Ins(1,3,4,5)*P*_4_ together with 50mM unlabeled Ins(1,3,4,5)*P*_4_ for 20 min, and the levels of ^3^H-inositol polyphosphates within the cells and in the extracellular medium were measured by HPLC [62]. This analysis not only confirmed incorporation of ^3^H-Ins(1,3,4,5)*P*_4_ but it also indicated that ~45% of this compound was converted into distinct metabolic products in the timeframe of the experiment [62]. We later confirmed that incubation of SKOV-3 cells with ^32^P-Ins*P*_5_ for different times resulted in an efficient and time-dependent incorporation of the radiolabeled compounds, as assessed by HPLC analysis [64]. Moreover, we observed a slow turnover of Ins*P*_5_ with only 5.0% of the total being converted into different metabolites after 30 min of incubation and only 6.2% converted after 1 h of incubation. The slow turnover of Ins*P*_5_ was confirmed by pulse-chase experiments in SKOV-3 cells that revealed that 84.6% of total ^3^H-Ins*P*_5_ but only 7% of total ^3^H-Ins(1,3,4,5)*P*_4_ was still detectable intracellularly after 5 h [37]. The mechanism of inositol polyphosphate incorporation is still not completely understood. A study investigating internalization of exogenous Ins*P*_6_ in HeLa cells proposed it to occur via pinocytosis [69]. A transporter was also identified in *Arabidopsis* [70], but further studies are required to establish the precise mechanism of incorporation/uptake.

During our studies, we observed that uptake of inositol polyphosphates differed between cell lines (Maffucci and Falasca, unpublished observations). This observation led us to hypothesize that increasing Ins*P*_5_ intracellular uptake could represent an alternative strategy to potentiate its activity. Indeed, some preliminary data in our laboratory indicated that a hydrophobic pro-drug of Ins*P*_5_ (Bt- Ins*P*_5_ PM, SiChem, Bremen, Germany), which could enter cells by diffusion, showed increased activity (as assessed by cell counting after 72h treatment) compared to the parental molecule (Maffucci and Falasca, unpublished observations). This was observed not only in cells sensitive to Ins*P*_5_, such as SKOV-3 and SKBR3 (IC_50_ 4.5 and 4.8 μM for Bt- Ins*P*_5_ vs. 23 and 21 μM for 2-*O*-Bn-Ins*P*_5_ and <50 μM for Ins*P*_5_), but also in cells more resistant to treatment with Ins*P*_5_, such as PC3, ASPC1, PANC1, and MDA-MB-468 cells (IC_50_ 27.6, 14.7, 11.2, and 16.9 μM, respectively vs. IC_50_ <50 μM; Maffucci and Falasca, unpublished observations). These preliminary observations suggest that the anti-cancer activity of Ins*P*_5_ can be greatly increased by improving its intracellular delivery.

### 5.3. Nanodelivery

Efficient delivery within the cell is one of the main issues with inositol polyphosphates or inositol polyphosphate analogues. In this respect, nanomedicines, which can target drugs at the site of action, might represent an interesting and valuable strategy. The availability of delivery systems provides an unprecedented opportunity to encapsulate variety of drugs, including negatively charged molecules, with high loading capacity and improved oral delivery. Therefore, it would be interesting to explore the possibility that nanodelivery strategies might improve the therapeutic potential of inositol polyphosphates, as it has been successfully achieved for other Akt/PDK1 inhibitors [71,72].

## 6. Targeting PH Domains of Signaling Proteins: Beyond Akt

Although the interest on PH domain inhibitors has been mainly polarized by Akt PH domain, other PH domain-containing signaling proteins have been shown to be potential targets. As mentioned before, 2-*O*-Bn-Ins*P*_5_ binds to PDK1 PH domain and inhibits the assembly of a protein complex between PLCγ1 and PDK1. More recently, we found that 2-*O*-Bn-Ins*P*_5_ inhibits 2D and 3D growth of different pancreatic cancer cell lines but not 2D growth of normal pancreatic cells, and it potentiates the activity of inhibitors of the PI3K isoform p110γ [73]. In addition, it has been found that introduction of WT-*TP53* increased the sensitivity of human pancreatic cancer cell line MIA-PaCa-2 to 2-*O*-Bn-Ins*P*_5_ [74]. Subsequently, it has been shown that β-estradiol strongly potentiated 2-*O*-Bn-Ins*P*_5_ efficacy to inhibit the growth of human pancreatic cancer cell lines BxPC3 and MIA-PaCa-2 [75]. Taking into consideration the potential role of PDK1 in the development of chemoresistance in different cancer types, this evidence suggests that targeting PDK1 PH domain promotes chemosensitization [76]. Bruton ‘s tyrosine kinase (Btk) and Interleukin-2-inducible T cell kinase (Itk) are known to play a key role in B cell and T cell receptor signaling, respectively [77]. Both Btk and Itk are localized in the cytoplasm in resting cells and translocate to the plasma membrane upon receptor activation through binding of their PH domains to PtdIns(3,4,5)*P*_3_ [78]. Small compounds targeting the interaction between Btk PH domain and PtdIns(3,4,5)*P*_3_ have been developed [79]. Interestingly, inositol polyphosphates have been found to regulate both Btk and Itk through PH domain binding [79,80,81]. Similarly, compounds targeting the PH domain of the PtdIns(3,4,5)*P*_3_-dependent Rac exchanger 1 have been recently developed and tested in vitro and in vivo, demonstrating that this PH domain is a drug target [82]. Strategies using in silico screening of a library of compounds to identify possible PH domain ligands have been recently developed. A successful example is the identification of a PH domain-binding molecule (PHT-7.3) as a candidate inhibitor of the scaffold protein connector enhancer of kinase suppressor of Ras 1 (Cnk1) through molecular modelling. Remarkably, PHT-7.3 has been shown to inhibit Cnk1 plasma membrane co-localization with KRas mutant and to block KRas-dependent cell growth in vivo and in vitro [83].

## 7. Conclusions

Over the past 20 years, our work has contributed to establishing inositol polyphosphates as promising leading compounds to design Akt inhibitors that specifically prevent its PtdIns(3,4,5)*P*_3_-dependent plasma membrane translocation. Our studies not only identified the anti-cancer properties of Ins*P*_5_, but they also indicated strategies to improve its activity. Our demonstration that chemical modifications can lead to compounds with increased selectivity towards other components of the PI3K-dependent signaling pathways represents an important step forwards for development of more specific inhibitors that might overcome limitations due to potential non-specific effects of inositol polyphosphates. On the other hand, our data also indicate that strategies that increase their intracellular uptake or delivery (chemical modification and/or complex formation) might result in more efficient anti-cancer activity. In this respect, a better understanding of the mechanisms of inositol polyphosphate uptake would provide crucial information to exploit their therapeutic potential fully.

In addition, structural biology can provide crucial insight, currently not fully appreciated, in the interactions between inositol polyphosphates and PH domains that can lead to the discovery of novel drug candidates [84]. In this respect, the combination of crystallography, molecular modelling, mutagenesis, structure-activity relationship and membrane-binding assays [85] can provide novel opportunity in the development of drugs specifically targeting the PH domain–phosphoinositides interaction.

## Figures and Tables

**Figure 1 ijms-21-07198-f001:**
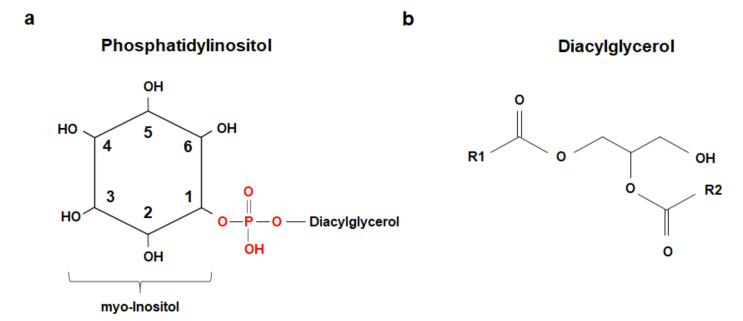
(**a**) Schematic representation of Phosphatidylinositol, showing the unphosphorylated hydroxyls within the myo-inositol head group, the phosphodiester bond (in red), and the attached diacylglycerol. (**b**) Generic structure of a diacylglycerol. Most phosphoinositides possess stearoyl (R1) and arachidonoyl (R2) chains.

**Figure 2 ijms-21-07198-f002:**
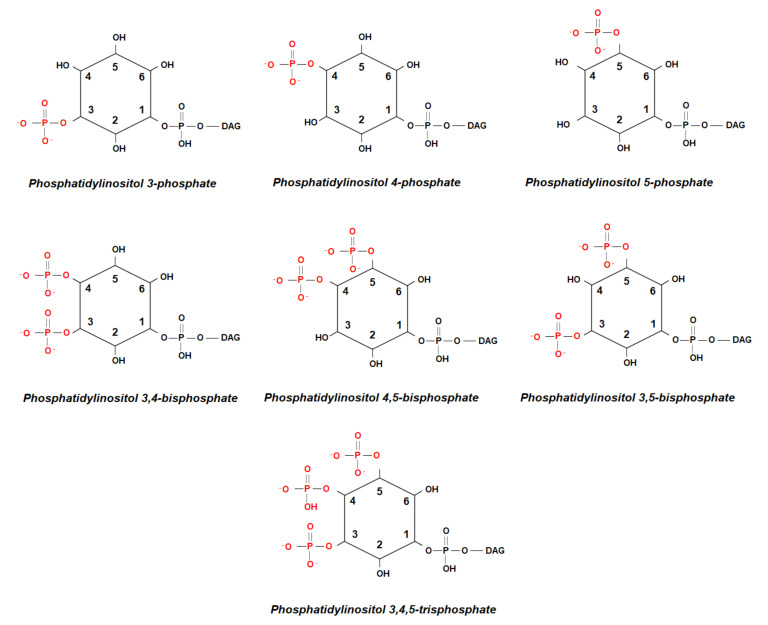
Schematic representation of the seven phosphoinositides.

**Figure 3 ijms-21-07198-f003:**
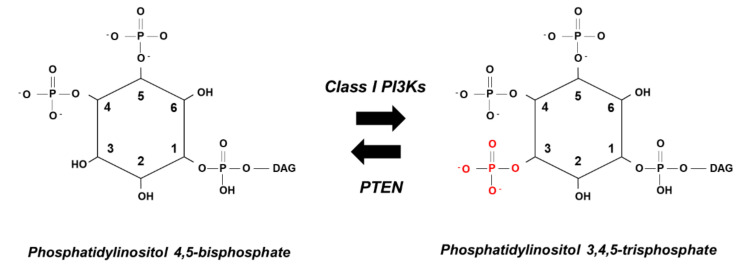
Schematic representation of the opposite action of class I PI3Ks and PTEN.

**Figure 4 ijms-21-07198-f004:**
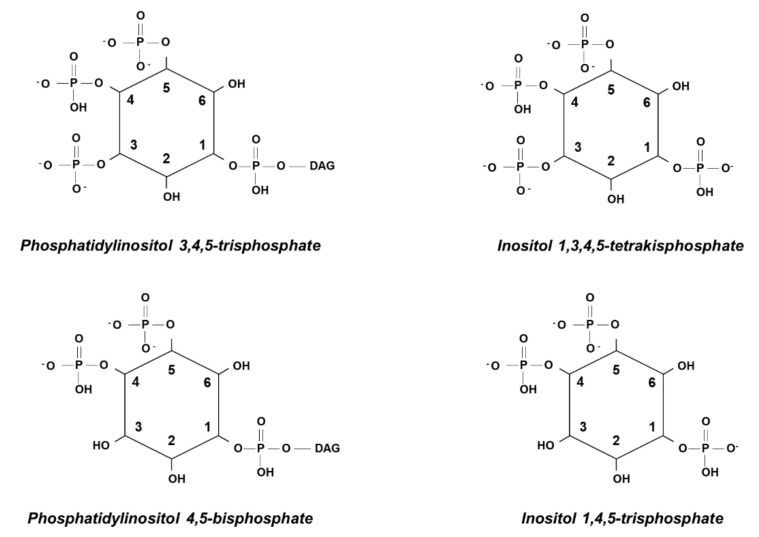
Schematic representation of phosphoinositides and their corresponding inositol polyphosphates.

**Figure 5 ijms-21-07198-f005:**
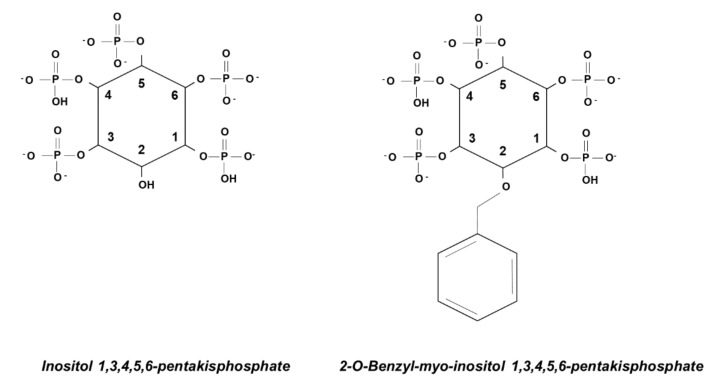
Schematic representation of Ins*P*_5_ and its derivative 2-*O*-Bn-Ins*P*_5_.

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
