# Peer review of "Inositol Polyphosphate-Based Compounds as Inhibitors of Phosphoinositide 3-Kinase-Dependent Signaling"

_ijms, 2020, doi:10.3390/ijms21197198_

Round 1
Reviewer 1 Report
Authors successfully reviewed the value and significance of IPs in the fine control of PI3K signaling. The only concern that I found is the ignorance of the following paper showing that 5-IP7 acts as a high-affinity inositol phosphate to inhibit Akt via binding to its PH domain. Authors should include the key discoveries of 5-IP7 actions in Akt control. Cell. 2010 Dec 10; 143(6): 897–910. Inositol pyrophosphates inhibit Akt signaling, regulate insulin sensitivity and weight gainAuthor Response
This is a good point and we have added the citation suggested
Reviewer 2 Report
This manuscript well descript about PI3K inhibitor compounds.
Author Response
We have revised the manuscript and corrected all grammar and spelling mistakes as suggested